# SPHERICAL CAUTIOUS OPTIMIZERS

**Jinghui Yuan,    Feiping Nie**[*]

School of Artificial Intelligence, Optics and Electronics (iOPEN),
Northwestern Polytechnical University,
Xi'an 710072, P.R. China
`yuanjh@mail.nwpu.edu.cn;feipingnie@gmail.com`

## ABSTRACT

Recent studies have shown that optimizing parameters with spherical constraints or scale-invariance can effectively improve model performance. This paper proposes the Spherical Cautious Optimizers. Standard cautious optimizers prevent overshooting by element-wise masking of updates with inconsistent signs. However, directly applying them to spherical or scale-invariant parameters allows radial noise to severely interfere with sign judgment, introducing geometric distortion and disrupting convergence. The Spherical Cautious Optimizers determine masks solely based on the sign consistency of updates and gradients in the tangent space, ensuring decisions are guided by the true feature learning directions, and employs retraction to align the optimization trajectory with the manifold geometry. Both theoretical analysis and experimental results show that the Spherical Cautious Optimizers guarantees monotonic descent while significantly improving convergence speed and accuracy in vision and language models. The method is highly general, requiring only a single line of code modification, and is applicable to general manifold constraints. We have released the code on the homepage[1].

## 1 INTRODUCTION

---

**Algorithm 1** *Spherical Cautious Optimizers*: $\alpha$ is the adaptive correction factor of the cautious optimizer. $\odot$ denotes element-wise multiplication, and $\mathbb{I}\{\cdot\}$ denotes the element-wise indicator function, which equals 1 if the condition is satisfied and 0 otherwise. $\mathcal{R}$ denotes the retraction operator, and $(\cdot)^\perp$ is the component in the tangent space at $w$.

---

**Require:** Parameter $w$, Update $u$ from Opt, Gradient $g$, Learning rate $\eta$, Decay rate $\lambda$

$\quad w \leftarrow w - \alpha\eta \cdot \mathbb{I}\{u \odot g > 0\} \odot u - \eta\lambda \cdot w$ ◆ Cautious Optimizer

$\quad w \leftarrow \mathcal{R}(w - \alpha\eta \cdot \mathbb{I}\{u^\perp \odot g^\perp > 0\} \odot u^\perp)$ ✓ Spherical Cautious Optimizer

---

Recently, Wen et al. (2025) and Xie et al. (2026) proposed performing optimization on the sphere and the spectral sphere, respectively, demonstrated effective improvements in model stability and generalization. Loshchilov et al. (2025) designed Transformer models with parameters lying on a hypersphere, accelerating the learning speed. Moreover, deep neural networks contain a large number of scale-invariant parameters (Salimans & Kingma, 2016), which can be viewed as a quotient manifold (Boumal, 2023) of the sphere. Therefore, how to efficiently optimize on the sphere has become an important direction for the future development of deep learning.

This paper aims to extend a highly effective optimizer plug-in in Euclidean space, the cautious optimizer (Liang et al., 2024) to the sphere. Cautious optimizers have been widely validated to mitigate momentum overshooting, accelerate the convergence of optimizers such as AdamW (Loshchilov & Hutter, 2019), Adan (Xie et al., 2024; Yuan et al., 2025b), and MARS (Yuan et al., 2025a), and have been integrated into well-known open-source libraries such as PyTorch Image Models (Wightman, 2019). Their core idea is to element-wise mask momentum components whose signs disagree with the gradients at each update, following the philosophy that *it's always better to be safe than sorry*.

---

[*]Corresponding author.

[1]https://github.com/Yuan-Jinghui/Spherical-Cautious-Optimizers

However, cautious optimizers are not suitable for spherically constrained or scale-invariant parameters. For spherically constrained or scale-invariant parameters, we care about directions rather than radial magnitudes (Wan et al., 2021). Since the update direction $u$ or the gradient $g$ contains a radial component, this component acts as noise that interferes with the mask's assessment of tangential feature alignment, severely affecting the stability and effectiveness of the learning process. Therefore, developing a cautious-optimizer plugin tailored to spherical optimization is key to achieving fast optimization of scale-invariant parameters.

To address this issue, we propose a spherical cautious optimizer plugin. All element-wise comparisons are performed in the tangent space, thereby avoiding interference from radial-component noise on the mask. We theoretically analyze the properties of the spherical cautious optimizer and validate its effectiveness on both language models and vision models. Experimental results show that, compared with applying the cautious optimizer mask, the spherical cautious optimizer can significantly improve model performance and convergence speed, demonstrating promising potential impact.

## 2 METHOD OF SPHERICAL CAUTIOUS OPTIMIZER

First, we briefly introduce the existing cautious optimizers. The update rule of the cautious optimizer is given in Equation (1), where $w$ denotes the model parameters, $g$ is the gradient, $\eta$ represents the learning rate, $\lambda$ is the weight decay coefficient, and $u$ denotes the proposed update. The operators $\odot$ and $\mathbb{I}\{\cdot\}$ indicate element-wise multiplication and element-wise indicator function (returning 1 if the condition is satisfied and 0 otherwise), respectively.

$$w \leftarrow w - \eta\alpha \cdot \mathbb{I}\{u \odot g > 0\} \odot u - \eta\lambda \cdot w \tag{1}$$

Here, $\mathbb{I}\{u \odot g > 0\} \odot u$ constitutes the core mechanism of the cautious optimizer: the components of the proposed update $u$ whose signs are consistent with the current gradient $g$ are regarded as trustworthy directions and are applied to update the parameters. In contrast, components with opposite signs are treated as directions that require further evaluation. They are accumulated in the momentum but are not used to update the parameters. The scalar $\alpha$ serves as a compensation factor after masking, defined as $\alpha = \frac{\sum_i \mathbb{I}\{u\}_i}{\sum_i \mathbb{I}\{u \odot g > 0\}_i + \epsilon}$.

However, when considering spherical constraints or scale-invariant parameters $w$, the above mask is affected by interference from radial noise. Specifically, for an update vector $u$, it can always be decomposed into a tangential component and a radial component, i.e., $u = u^\perp + u^\parallel$, where $\langle u^\perp, w \rangle = 0$. For spherical or scale-invariant parameters $w$, $u^\perp$ truly determines the direction of feature learning, while $u^\parallel$ only serves to increase or decrease the norm of $w$. In this case, $u^\parallel$ is pure noise, and similarly $g^\parallel$ is also noise. However, the cautious optimizer performs element-wise sign comparisons between $u^\perp + u^\parallel$ and $g^\perp + g^\parallel$ to obtain the mask. This artificially introduces the noise components $u^\parallel$ and $g^\parallel$, which interferes with model convergence.

Therefore, the most reasonable approach is to project both the gradient and the update onto the tangent space, filtering out noise and retaining the pure feature learning direction for element-wise comparison and mask application. The mask should then be defined as $\mathbb{I}\{u^\perp \odot g^\perp > 0\}$, and the parameter update is performed as shown in Equation (2). This idea is applicable not only to spheres but also to general manifolds (Yuan et al., 2026).

$$w \leftarrow \mathcal{R}(w - \alpha\eta \cdot \mathbb{I}\{u^\perp \odot g^\perp > 0\} \odot u^\perp) \tag{2}$$

Here, $\mathcal{R}$ denotes a retraction. For strict spherical constraints, one can choose $\mathcal{R}(\xi) = \xi/\|\xi\|$, or alternatively apply weight decay as an implicit spherical, i.e., $\mathcal{R}(\xi) = \xi - \eta\lambda \cdot w$. In particular, for the Euclidean sphere, $u^\perp$ and $g^\perp$ are computed according to Equation (3):

$$u^\perp = u - \frac{\langle u, w \rangle}{\|w\|^2}w, \quad g^\perp = g - \frac{\langle g, w \rangle}{\|w\|^2}w. \tag{3}$$

For the spectral sphere constraints, $u^\perp$ and $g^\perp$ are computed according to Equation (4), where $q_1$ and $v_1$ are the left and right leading singular vectors of $w$.

$$u^\perp = u - \langle u, q_1 v_1^T \rangle q_1 v_1^T, \quad g^\perp = g - \langle g, q_1 v_1^T \rangle q_1 v_1^T. \tag{4}$$

These modifications incur almost zero cost, because when optimizing parameters $w$ constrained on a sphere, it is usually necessary to compute the tangential components of the gradient or the update. We simply reuse these previously computed quantities to construct the mask, without increasing computational complexity, and likewise require only a single line of code.

## 3   THEORETICAL ANALYSIS

**Theorem 1.** *Define the Euclidean cautious mask $M^{\mathbb{R}}(u,g) = \mathbb{I}\{u \odot g > 0\}$, and the spherical cautious mask $M^{\mathbb{S}}(u,g) = \mathbb{I}\{u^{\perp} \odot g^{\perp} > 0\}$. For any nonzero radial perturbation $\delta \cdot w$, $M^{\mathbb{R}}$ may flip, $M^{\mathbb{S}}$ always remains unchanged. This property is robust to interference from radial noise.*

The proof of Theorem 1 is straightforward: all radial components are removed before constructing the mask, and therefore the mask is clearly independent of the radial component. Nevertheless, Theorem 1 serves as the core motivation for the spherical cautious optimizer. When $w$ is constrained on sphere or scale-invariant parameters, the radial component acts as noise and should not be retained.

**Theorem 2.** *Consider a smooth objective function $f : \mathbb{S} \to \mathbb{R}$ on the sphere with the retraction $\mathcal{R}$, tangential gradient $g_t^{\perp}$, and tangential update $u_t^{\perp}$. Using the naive Euclidean mask $M_t^{\mathbb{R}} = \mathbb{I}\{u_t \odot g_t > 0\}$ to filter tangential updates does not guarantee monotonic descent, as radial components may cause sign mismatch between ambient and tangent spaces. In contrast, under the Spherical Cautious Optimizers where $\xi_t = \alpha M_t^{\mathbb{S}} \odot u_t^{\perp}$ with the mask $M_t^{\mathbb{S}} = \mathbb{I}\{u_t^{\perp} \odot g_t^{\perp} > 0\}$, there exists a sufficiently small step size $\eta > 0$ such that $f(w_{t+1}) \leq f(w_t)$. The equality holds if and only if $w_t$ is a critical point or the update is fully masked.*

*Proof.* The Taylor expansion on the sphere gives $f(w_{t+1}) = f(w_t) - \eta \langle g_t^{\perp}, \xi_t \rangle + O(\eta^2)$. For the **Euclidean Mask**, the condition $(u_t)_i(g_t)_i > 0$ does not imply $(u_t^{\perp})_i(g_t^{\perp})_i > 0$ due to radial noise (e.g., strong radial components can flip the sign). Consequently, the summation $\alpha \sum_i (g_t^{\perp})_i (u_t^{\perp})_i M_{t,i}^{\mathbb{R}}$ may contain negative terms where the mask erroneously allows directions that oppose the gradient, leading to potential ascent $f(w_{t+1}) > f(w_t)$. For the **Spherical Mask**, the update direction is defined by strictly consistent tangential signs: $\langle g_t^{\perp}, \xi_t \rangle = \alpha \sum_i (g_t^{\perp})_i (u_t^{\perp})_i \cdot \mathbb{I}\{(u_t^{\perp})_i (g_t^{\perp})_i > 0\}$. This ensures every active term in the summation is non-negative, guaranteeing $\langle g_t^{\perp}, \xi_t \rangle \geq 0$. Since the descent direction is validated in the tangent space, the first-order term $-\eta \langle g_t^{\perp}, \xi_t \rangle$ remains non-positive and dominates the $O(\eta^2)$ error, ensuring strict non-increase. □

Theorem 2 shows that applying the spherical mask guarantees strict descent, whereas the Euclidean cautious mask cannot. In particular, for scale-invariant parameters, $g_t = g_t^{\perp}$. Although this guarantees that $\langle M_t^{\mathbb{R}} u_t, g_t \rangle > 0$, certain coordinates of $u_t$ may still be misled, resulting in incorrect masking decisions. This is inconsistent with the design philosophy of the cautious optimizer.

**Theorem 3.** *Assume the optimization stagnates at a non-critical point $w_t$ with Euclidean gradient $g_t$, where $\|g_t^{\perp}\| > 0$. Consider a momentum buffer updated in the ambient space: $m_t = \beta m_{t-1} + (1 - \beta)g_t$. Let the spherical cautious mask be determined by the $M_t^{\mathbb{S}} = \mathbb{I}\{m_t^{\perp} \odot g^{\perp} > 0\}$. Then, there exists a finite step $k$ such that the mask activates.*

*Proof.* Assume the optimization stagnates at step $t$ so that $w_{t+k} = w_t$ for all $k \geq 0$. Consequently, the ambient gradient $g_t$ remains constant. Let the ambient gradient be decomposed into tangential and radial components: $g_t = g_t^{\perp} + g_t^{\parallel}$. Under standard momentum dynamics (e.g., $m_t = \beta m_{t-1} + (1 - \beta)g_t$), the ambient momentum converges to the gradient as inputs remain constant: $\lim_{k \to \infty} m_{t+k} = g_t$. The Spherical Cautious Optimizer projects this accumulated momentum onto the tangent space for masking. Applying the linearity of the projection operator $\mathcal{P}_T(\cdot)$, we have

$$\lim_{k \to \infty} m_{t+k}^{\perp} = \mathcal{P}_T(\lim_{k \to \infty} m_{t+k}) = \mathcal{P}_T(g_t^{\perp} + g_t^{\parallel}) = g_t^{\perp} + 0 = g_t^{\perp} \tag{5}$$

Note that the radial component $g_t^{\parallel}$ is filtered out by the projection. Finally, we examine the mask condition element-wise. For the $i$-th parameter, $\lim_{k \to \infty} (m_{t+k}^{\perp})_i \cdot (g_t^{\perp})_i = (g_t^{\perp})_i^2 \geq 0$. Since $w_t$ is not a critical point, $\|g_t^{\perp}\| > 0$, implying there exists at least one index $j$ such that $(g_t^{\perp})_j \neq 0$, which yields a strictly positive product $(g_t^{\perp})_j^2 > 0$. Thus, the mask $M_t^{\mathbb{S}}$ will eventually activate. □

**It is worth noting that** the masked update $\alpha M_t^{\mathbb{S}} \odot u_t^{\perp}$ may slightly deviate from the tangent space. Therefore, if one wishes the update to remain along the tangent direction, a projection operation can be applied optionally. Since $\langle \alpha M_t^{\mathbb{S}} \odot u_t^{\perp}, g_t^{\perp} \rangle = \langle (\alpha M_t^{\mathbb{S}} \odot u_t^{\perp})^{\perp}, g_t^{\perp} \rangle$, descent is still preserved.

Another alternative feasible choice is to apply a mask to the entire vector, i.e., a vector-wise masked update $\mathbb{I}\{\langle g^{\perp}, u^{\perp} \rangle > 0\}u^{\perp}$. However, such a vector mask may cause the update to completely vanish at some steps. This represents a trade-off.

Table 1: Final Training and Test Loss across different learning rates on NLP task

| Method | LR = 0.001 | | LR = 0.0015 | | LR = 0.002 | | LR = 0.0025 | |
|--------|-------|------|-------|------|-------|------|-------|------|
| | Train | Test | Train | Test | Train | Test | Train | Test |
| AdamP | 5.7557 | 5.9773 | 6.0742 | 6.2591 | 6.2693 | 6.4625 | 6.4368 | 6.6477 |
| AdamW | 5.8861 | 6.1111 | 6.1434 | 6.3701 | 6.4926 | 6.6487 | 6.5767 | 6.7954 |
| C-AdamP | 5.8557 | 6.0897 | 6.2830 | 6.5053 | 6.3577 | 6.5983 | 6.4275 | 6.6416 |
| C-AdamW | 5.7945 | 6.0236 | 6.1669 | 6.3890 | 6.2820 | 6.5091 | 6.3925 | 6.5991 |
| SC-AdamW | 5.7929 | 6.0178 | 6.0685 | 6.3045 | 6.3065 | 6.5184 | 6.3944 | 6.6619 |
| **SC-AdamP** | **5.7030** | **5.9398** | **5.8686** | **6.1153** | **5.9914** | **6.2230** | **6.0547** | **6.2934** |

## 4 EXPERIMENTS

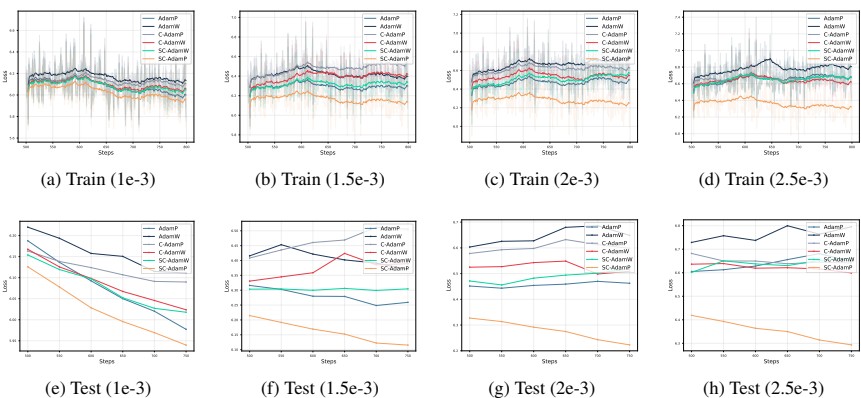

(a) Train (1e-3)     (b) Train (1.5e-3)     (c) Train (2e-3)     (d) Train (2.5e-3)

(e) Test (1e-3)     (f) Test (1.5e-3)     (g) Test (2e-3)     (h) Test (2.5e-3)

Figure 1: Training and test loss curves under different learning rates.

To validate the optimizers, we take AdamP (Heo et al., 2020) as an example and apply the spherical cautious optimizers to scale-invariant parameters, while other parameters are optimized using the cautious optimizers, $\mathcal{R}(\xi) = \xi - \eta\lambda \cdot w$. We evaluate the methods on both NLP and CV tasks. The compared algorithms include AdamP (Heo et al., 2020), AdamW, Cautious AdamW (C-AdamW), Cautious AdamP (C-AdamP), Spherical Cautious AdamW (SC-AdamW), in which all parameters

Table 2: Best Test accuracy on CV task

| Method | WD=0 | WD Act. |
|--------|------|---------|
| AdamP | 93.33 | 93.47 |
| AdamW | 92.90 | 92.99 |
| C-AdamP | 93.37 | 93.68 |
| C-AdamW | 93.25 | 93.14 |
| **SC-AdamP** | **93.82** | **93.77** |

use the spherical cautious mask and Spherical Cautious AdamP (SC-AdamP, Ours).

For the NLP task, we choose the 0.5B Qwen (Bai et al., 2023) model and use a subset of OpenWebText-100k for one epoch, with batch size 16 and 800 training steps. The learning rate is selected from $\{1, 1.5, 2, 2.5\} \times 10^{-3}$. For the AdamP family, the weight decay is set to 0, while for the AdamW family, the weight decay is set to 0.1. For the CV task, we choose ResNet-18 (He et al., 2016) with Cifar-10 and train for 70 epochs with batch size 1024 and learning rate $3 \times 10^{-3}$. We evaluate two settings: (1). All optimizers with weight decay set to 0. (2). AdamP family with weight decay set to 0.01, while the AdamW family of optimizers uses a weight decay of 0.1.

Table 1 reports the final-step training loss and test loss for the NLP task, while Figure 2 shows the training and test loss curves for the NLP task. Table 2 presents the best validation accuracy for the CV task. All these results indicate that when parameters are constrained on or near a sphere, spherical cautious optimizers can bring significant improvements, whereas applying Euclidean cautious optimizers may instead introduce noise that interferes with learning.

**Overall**, compared with the standard Euclidean Cautious Optimizer, we propose a Spherical Cautious Optimizer plugin that is tailored for parameters constrained on or near a sphere. By effectively filtering out radial noise, our method significantly improves optimization efficiency.

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

# A    RUNNING AND REFERENCE CODE

## A.1    RUN QUICKLY

The PyTorch Image Models library[2] (timm) now supports the use of the spherical cautious optimizers. To use it quickly, please first download the timm library.

```
pip install timm
```

Then, please set up the Cifar-10[3] dataset yourself and run:

```
DATA_DIR=/home/data
NUM_GPUS=1

./distributed_train.sh $NUM_GPUS $DATA_DIR \
    --dataset torch/cifar10 \
    --model resnet18 \
    --input-size 3 224 224 \
    --num-classes 10 \
    --batch-size 1024 \
    --epochs 70 \
    --opt cadamp \
    --lr 3e-3 \
    --weight-decay 0 \
    --amp \
    --experiment quick_cadamp_test
```

Please note that cadamp in timm corresponds to SC-AdamP in our paper. For naming consistency, it is called cadamp in timm. It applies the spherical optimizer masks to scale-invariant parameters, while standard cautious optimizer masks are used for regular parameters.

## A.2    REFERENCE CODE

```python
import torch
import torch.nn.functional as F
from torch.optim.optimizer import Optimizer
import math

def _channel_view(x) -> torch.Tensor:
    return x.reshape(x.size(0), -1)

def _layer_view(x) -> torch.Tensor:
    return x.reshape(1, -1)

def projection(p, grad, perturb, delta: float, wd_ratio: float, eps:
    float, caution: bool = False):
    wd = 1.
    expand_size = (-1,) + (1,) * (len(p.shape) - 1)
    for view_func in [_channel_view, _layer_view]:
        param_view = view_func(p)
        grad_view = view_func(grad)
        cosine_sim = F.cosine_similarity(grad_view, param_view, dim=1,
            eps=eps).abs_()

        # FIXME this is a problem for PyTorch XLA
        if cosine_sim.max() < delta / math.sqrt(param_view.size(1)):
            p_n = p / param_view.norm(p=2, dim=1).add_(eps).reshape(
                expand_size)
```

---

[2]https://github.com/huggingface/pytorch-image-models
[3]https://www.cs.toronto.edu/ kriz/cifar.html

```python
                    perturb -= p_n * view_func(p_n * perturb).sum(dim=1).reshape(
                        expand_size)

                    if caution:
                        # Spherical Cautious Optimizer Logic
                        grad_radial = p_n * view_func(p_n * grad).sum(dim=1).
                            reshape(expand_size)
                        grad_perp = grad - grad_radial

                        mask = (perturb * grad_perp > 0).to(grad.dtype)
                        mask.div_(mask.mean().clamp_(min=1e-3))
                        perturb.mul_(mask)
                        # Enhance the numerical stability of the Caution
                        perturb -= p_n * view_func(p_n * perturb).sum(dim=1).
                            reshape(expand_size)
                    wd = wd_ratio
                    return perturb, wd

            if caution:
                # Standard Cautious Optimizer Logic for non-projected parameters
                mask = (perturb * grad > 0).to(grad.dtype)
                mask.div_(mask.mean().clamp_(min=1e-3))
                perturb.mul_(mask)

            return perturb, wd

class AdamP(Optimizer):
    def __init__(
            self,
            params,
            lr=1e-3,
            betas=(0.9, 0.999),
            eps=1e-8,
            weight_decay=0,
            delta=0.1,
            wd_ratio=0.1,
            nesterov=False,
            caution=False,
    ):
        defaults = dict(
            lr=lr,
            betas=betas,
            eps=eps,
            weight_decay=weight_decay,
            delta=delta,
            wd_ratio=wd_ratio,
            nesterov=nesterov,
            caution=caution,
        )
        super(AdamP, self).__init__(params, defaults)

    @torch.no_grad()
    def step(self, closure=None):
        loss = None
        if closure is not None:
            with torch.enable_grad():
                loss = closure()

        for group in self.param_groups:
            for p in group['params']:
                if p.grad is None:
                    continue

                grad = p.grad
```

```python
            beta1, beta2 = group['betas']
            nesterov = group['nesterov']
            caution = group.get('caution', False)

            state = self.state[p]

            # State initialization
            if len(state) == 0:
                state['step'] = 0
                state['exp_avg'] = torch.zeros_like(p)
                state['exp_avg_sq'] = torch.zeros_like(p)

            # Adam
            exp_avg, exp_avg_sq = state['exp_avg'], state['exp_avg_sq
                ']

            state['step'] += 1
            bias_correction1 = 1 - beta1 ** state['step']
            bias_correction2 = 1 - beta2 ** state['step']

            exp_avg.mul_(beta1).add_(grad, alpha=1 - beta1)
            exp_avg_sq.mul_(beta2).addcmul_(grad, grad, value=1 -
                beta2)

            denom = (exp_avg_sq.sqrt() / math.sqrt(bias_correction2))
                .add_(group['eps'])
            step_size = group['lr'] / bias_correction1

            if nesterov:
                perturb = (beta1 * exp_avg + (1 - beta1) * grad) /
                    denom
            else:
                perturb = exp_avg / denom

            # Projection
            wd_ratio = 1.
            if len(p.shape) > 1:
                perturb, wd_ratio = projection(
                    p, grad, perturb, group['delta'], group['wd_ratio
                        '], group['eps'], caution
                )
            elif caution:
                # Apply standard caution for scalars/1D tensors if
                    needed
                mask = (perturb * grad > 0).to(grad.dtype)
                mask.div_(mask.mean().clamp_(min=1e-3))
                perturb.mul_(mask)

            # Weight decay
            if group['weight_decay'] > 0:
                p.mul_(1. - group['lr'] * group['weight_decay'] *
                    wd_ratio)

            # Step
            p.add_(perturb, alpha=-step_size)

    return loss
```

In addition, the SGDP optimizer equipped with the spherical cautious optimizers has also been supported in the timm library. It can be imported using a similar approach, or the detailed code can be found directly on the timm official website[4].

---

[4]https://github.com/huggingface/pytorch-image-models/blob/main/timm/optim/sgdp.py

## B    HARD HYPERSPHERICAL CONSTRAINT

In addition, we also tested the results of normalizing scale-invariant parameters onto the hypersphere instead of using implicit weight decay. Specifically, we trained ResNet-18 on mini-ImageNet[5] while keeping all parameters, including the random seed, exactly the same. The code used for the experiment is as follows:

```
DATA_DIR=/home/data
NUM_GPUS=2

./distributed_train.sh $NUM_GPUS $DATA_PATH \
    --num-classes 100 \
    --model resnet18 \
    -j 8 \
    --epochs 200 \
    --warmup-prefix \
    --sched-on-updates \
    --warmup-lr 0 \
    --mixup .2 \
    --model-ema \
    --model-ema-decay 0.999 \
    --model-ema-warmup \
    --aa rand-m9-mstd0.5-inc1 \
    --remode pixel \
    --reprob 0.25 \
    --amp \
    --weight-decay .05 \
    --drop 0.1 \
    --drop-path .1 \
    -b 288 \
    --opt $opt \
    --lr 1e-3 \
    --seed 1 \
    --experiment  run_${opt}_seed1
```

Compared with weight decay, spherical regularization normalizes scale-invariant parameters onto the hypersphere. Specifically, after each update, the parameters are divided by the new norm and then multiplied by the norm before the update, $w_{t+1} = \|w_t\| \frac{w_{t+1}}{\|w_{t+1}\|}$.

We find that in the final stage, spherical regularization can significantly improve the training results compared with weight decay. We preliminarily believe this is because even when the last step projects onto the tangent space of $w$ and applies weight decay, the norm can still grow slowly, which leads to a reduction in the effective learning rate of the scale-invariant parameters.

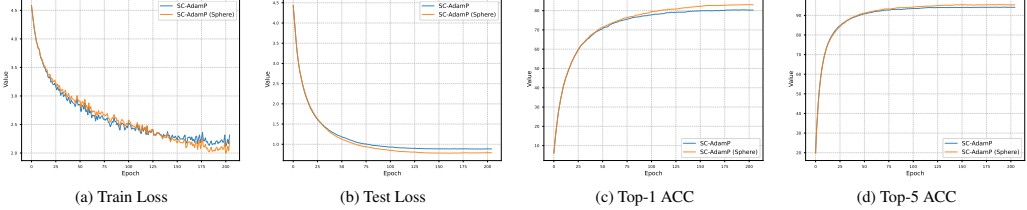

|     (a) Train Loss     |     (b) Test Loss     |     (c) Top-1 ACC     |     (d) Top-5 ACC     |

Figure 2: Comparison of results between spherical regularization and weight decay.

As scale-invariant parameters become increasingly widely used in deep learning, we believe it is necessary to design specialized optimization techniques for them, such as spherical cautious optimizers. We also look forward to working with the deep learning and optimization communities to explore how more advanced optimizers, such as Muon (Jordan et al., 2024), can be incorporated into the framework of spherical cautious optimization.

---

[5]https://huggingface.co/datasets/timm/mini-imagenet

## C    HOUSEHOLDER TRANSFORMATION

As pointed out, directly applying a mask in the tangent space is coordinate-dependent and may cause the masked vector to slightly deviate from the tangent space. In this section, we adopt the Hausdorff transformation to provide a method that is somewhat more computationally involved but better aligned with the underlying geometry.

For a spherical parameter $w$, when $w$ points to the north pole, the tangent space aligns with the coordinate axes of the ambient space. In this case, applying an element-wise mask ensures that the result remains in the tangent space. Let the north pole direction be $e_1 = (1, 0, \ldots, 0)$.

Our core idea is to use the Hausdorff transformation to map $w$ to the north pole direction. Then, we apply an element-wise mask to the transformed $u^\perp$ and $g^\perp$, and finally map them back. Specifically, we construct the deviation vector $v = w - e_1$, and then form the Hausdorff matrix $H = I - 2\frac{vv^T}{\|v\|^2}$. We map $g^\perp$ and $u^\perp$ to the north pole as $\tilde{u}^\perp = Hu^\perp, \tilde{g}^\perp = Hg^\perp$, where:

$$\tilde{u}^\perp = (0, \tilde{u}_2, \tilde{u}_3, \ldots, \tilde{u}_n), \quad \tilde{g}^\perp = (0, \tilde{g}_2, \tilde{g}_3, \ldots, \tilde{g}_n). \tag{6}$$

Subsequently, we apply an element-wise mask to $\tilde{u}^\perp$ and $\tilde{g}^\perp$, and then map the masked $\tilde{u}^\perp_{\text{mask}}$ back via the inverse Hausdorff transformation: $u^\perp_{\text{mask}} = H^{-1}\tilde{u}^\perp_{\text{mask}}$. Finally, $u^\perp_{\text{mask}}$ is used for the update.

This approach incurs a slightly higher computational cost, but is more consistent with the underlying geometry. For large-scale deep networks operating in noisy environments, the Hausdorff transformation is not always necessary. However, for deterministic Riemannian optimization, we still recommend using the Hausdorff transformation to obtain a more geometrically faithful masking.

