# OpenReview forum: "Spherical Cautious Optimizers"
_ICLR.cc/2026/Workshop/Sci4DL — Sci4DL 2026_

### Official Review · Reviewer_S638 · 2026-02-19

**Fit:** 3
**Significance:** 2
**Confidence:** 2

**Summary:**

This paper extends cautious optimizers to spherical and scale-invariant parameter settings. The authors propose constructing the mask using tangential components only and performing updates via retraction. Theoretical analysis establishes robustness to radial perturbations and first-order monotonic descent guarantees. Empirical results on language and vision models show consistent improvements in loss and accuracy over AdamW, AdamP, and their cautious variants.

**Strengths:**

- The paper identifies an inconsistency when applying Euclidean cautious masking to spherical or scale-invariant parameters, and the proposed modification is conceptually clean and well aligned with manifold optimization principles.

- Theoretical results are clearly stated and directly tied to the algorithmic motivation, particularly the robustness to radial perturbations and the monotonic descent guarantee.

- The method is lightweight and practical, requiring minimal code modification and incurring negligible computational overhead, as emphasized in both the algorithm description and reference implementation.

**Suggestions:**

- Parts of the theorems can be made more precise, for example, "This property is robust to interference from radial noise." Also, in Theorem 1, it is not clear whether "$M^\mathbb{R}$ may flip" means all elements in $M^\mathbb{R}$ change or at least one element changes.

- Performance gains, while consistent, are modest in magnitude.

---

### Official Review · Reviewer_JZ7F · 2026-02-27

**Fit:** 3
**Significance:** 2
**Confidence:** 2

**Summary:**

This paper presents the Spherical Cautious Optimizer, which is a extension of the Cautious Optimizer designed for parameters constrained on a sphere or exhibiting scale-invariance. The key idea is to perform element-wise sign comparisons in the tangent space rather than in the ambient Euclidean space. The proposed approach preserves the core advantage of cautious optimization while making it compatible with spherical optimization settings by removing noisy radial components. The method is simple to implement and can be adapted to existing optimizers with minimal modification.

**Strengths:**

- The proposed modification is conceptually simple and broadly applicable to existing cautious optimizers.
- The method requires only minor modifications to existing optimizers and introduces negligible additional computational cost.
- The authors provide clear theoretical arguments supporting their motivation and justifying the proposed approach.

**Suggestions:**

- Providing precise information about the experiments would improve reproducibility and clarity. The description of the experiments lacks some important details. For example:
    - The NLP task appears to be language modeling, and the reported loss seems to be perplexity, but this is not explicitly stated.
    - Based on the code in appendix, the CV task seems to use CIFAR-10, but this should be clearly mentioned in the main text or table captions.
- In Figure 1, using a single shared legend and simplifying the layout would improve readability and enhance the overall presentation quality, since all subfigures in Figure 1 currently use the same legends.
- It would strengthen the paper to include experiments on more practically relevant models (e.g., 3B-7B LLMs).

---

### Official Review · Reviewer_Hzs4 · 2026-02-27

**Fit:** 2
**Significance:** 2
**Confidence:** 1

**Summary:**

This work extends the Cautious Optimizers framework to spherically constrained and scale-invariant parameters. The authors observe that when the standard (Euclidean) cautious mask is applied on the sphere, the radial component of the gradient/update acts as noise that can corrupt the element-wise sign comparison. The fix is to project both gradient and update into the tangent space before masking. The paper provides theoretical guarantees and empirical validation showing modest improvements over the standard Euclidean cautious baselines (the practical significance of these gains is difficult to fully assess given the narrow experimental setting and lack of variance estimates). It is nice that the method adds negligible overhead, though the paper feels quite dependent on the preceding Cautious Optimizers work. However, I believe it is a well-scoped workshop contribution

**Strengths:**

- The failure mode is genuine, and the fix is meaningful. Theorem 2 shows the Euclidean mask can cause ascent on the sphere while the spherical mask guarantees strict descent. The tangent-space projection is the natural geometric correction.
- Practical relevance beyond explicit spherical constraints. Since normalization layers (BatchNorm, LayerNorm) make a large fraction of parameters in modern networks effectively scale-invariant, this fix is broadly applicable even though the paper frames it narrowly.

**Suggestions:**

- Adding variance estimates to empirical results would help the reader better assess the practical significance of these improvements.
- projection vs. masking ablation. It would be useful to see a baseline that just projects into the tangent space without applying the cautious mask. Right now it's unclear how much of the improvement comes from the projection alone (which is standard practice) vs. the mask interaction.
- Generality claim -- the paper says it applies to general manifold constraints, but only shows results on two types of spheres. This should be either a softened claim or provide better support.
- The descent guarantee needs "sufficiently small \eta" but there's no indication of how small that is in practice. Even a rough empirical comment would help.

---

### Meta-Review · Area_Chair_JjFE · 2026-03-01

**Recommendation:** Accept

**Metareview:**

I recommend an accept and suggest the authors to look at the reviewers suggestions.

---

### Decision · Program_Chairs · 2026-03-02

Accept